# Utility of CK8, CK10, CK13, and CK17 in Differential Diagnostics of Benign Lesions, Laryngeal Dysplasia, and Laryngeal Squamous Cell Carcinoma

**DOI:** 10.3390/diagnostics12123203

**Published:** 2022-12-16

**Authors:** Novica Boricic, Ivan Boricic, Ivan Soldatovic, Jovica Milovanovic, Aleksandar Trivic, Tatjana Terzic

**Affiliations:** 1Institute of Pathology, Faculty of Medicine, University of Belgrade, 11000 Belgrade, Serbia; 2Faculty of Medicine, Institute of Medical Statistics and Informatics, University of Belgrade, 11000 Belgrade, Serbia; 3Clinic for Otorhinolaryngology and Maxillofacial Surgery, Clinical Centre Serbia, Faculty of Medicine, University of Belgrade, 11000 Belgrade, Serbia

**Keywords:** cytokeratins, laryngeal carcinoma, laryngeal dysplasia, laryngeal polyp, laryngeal papilloma

## Abstract

There are no reliable immunohistochemical markers for diagnosing laryngeal squamous cell carcinoma (SCC) or diagnosing and grading laryngeal dysplasia. We aimed to evaluate the diagnostic utility of CK8, CK10, CK13, and CK17 in benign laryngeal lesions, laryngeal dysplasia, and laryngeal SCC. This retrospective study included 151 patients diagnosed with laryngeal papilloma, laryngeal polyps, laryngeal dysplasia, and laryngeal SCC who underwent surgical treatment between 2010 and 2020. Immunohistochemistry (IHC) was carried out using specific monoclonal antibodies against CK8, CK10, CK13, and CK17. Two experienced pathologists performed semi-quantitative scoring of IHC positivity. The diagnostic significance of the markers was analyzed. CK13 showed a sensitivity of 100% and a specificity of 82.5% for distinguishing between laryngeal SCC and laryngeal dysplasia and benign lesions. CK17 showed a sensitivity of 78.3% and specificity of 57.1% for the detection of laryngeal SCC vs. laryngeal dysplasia. CK10 showed a sensitivity of 80.0% for discriminating between low-grade and high-grade dysplasia, and a specificity of 61.1%. Loss of CK13 expression is a reliable diagnostic tool for diagnosing laryngeal lesions with malignant potential and determining resection lines. In lesions with diminished CK13 expression, CK17 could be used as an auxiliary immunohistochemical marker in diagnosing laryngeal SCC. In CK13-negative and CK17-positive lesions, CK10 positivity could be used to determine low-grade dysplasia. CK8 is not a useful IHC marker in differentiating between benign laryngeal lesions, laryngeal dysplasia, and laryngeal SCC.

## 1. Introduction

Laryngeal squamous cell carcinoma (SCC) is the second most common malignant tumor of the respiratory tract, just behind lung cancer, and represents one-fifth of all malignant tumors of the head and neck region. The global incidence of laryngeal SCC is 4.1 per 100,000 men and 0.63 per 100,000 women [1]. The diagnosis of laryngeal SCC is based on morphological characteristics using hematoxylin—eosin (H&E) staining. However, there are diagnostically very challenging cases, especially those involving biopsy specimens with limited tissue. Laryngeal dysplasia is a potentially malignant lesion of the laryngeal epithelia that has the potential to progress into SCC. The incidence of laryngeal dysplasia is 10.2 and 2.1 per 100,000 males and females, respectively [2]. Diagnosis of laryngeal dysplasia is also made based on morphological characteristics. There have been numerous classifications of laryngeal dysplasia, but all displayed significant interobserver variability [2]. The latest WHO 2017 classification was changed to incorporate all previous classifications to better correlate between the histological findings and biological behavior of the lesions. Unfortunately, there remains great interobserver variability for the WHO 2017 classification of laryngeal dysplasia [3].

It is necessary to make a reliable and accurate diagnosis of either malignant or premalignant lesions because the therapeutic approaches differ significantly, as well as the quality of life after therapy. To our knowledge, no immunohistochemical markers are reliable for diagnosing laryngeal SCC or diagnosing and grading laryngeal dysplasia [1]. Reliable IHC markers should be used for accurate diagnosis, monitoring histologic progression, and determining prognosis.

Cytokeratins are a group of intermediate filament proteins primarily responsible for retaining cellular mechanical and structural integrity. Cytokeratins have important roles in cell cycle control, cellular differentiation, and growth, and are involved in the process of apoptosis [4]. Several studies have shown that CK17 acts as a promoter of tumorigenesis [5]. The expression of CK17 has been investigated in laryngeal epithelium and laryngeal lesions, and CK17 emerged as a potential IHC marker for laryngeal SCC [6,7]. Downregulation of CK13 was noted in laryngeal SCC in one study [8]. One small study showed increased mRNA expression of CK8, CK10, and CK17 in laryngeal SCC [9]. In only one study to date, expressions of CK10, CK13, and CK8 have been investigated in benign vocal cord lesions, but with a sample size of only 12 cases. Loss of CK13 expression was noted in hyperkeratotic epithelia, a gain of CK10 expression was described in lesions with keratinization, and no expression of CK8 was noted [10]. To our knowledge, CK17 expression has been described in only six cases of vocal cord polyps [6].

It is presumed that the CK13 gene might be a tumor-suppressor gene that may play an important role in laryngeal cancerogenesis [8]. Proteomic and genomic studies suggest that CK17 is expressed in many malignant tumors [11,12]. In oral cancers, over-expression of CK17 promotes cell proliferation and migration by stimulating the Akt/mTOR pathway [13]. It is shown that KRT17 promotes the proliferation and invasion of skin tumors through the KRT17/HNRNP-K/CXCR3 pathway [14]. It is speculated that the function of K1/K10 expression might be a suppression of cell death [15]. In hepatocytes, CK8 modulates apoptosis induced by Fas [16], and the interaction of 14-3-3 family adapter proteins with K8 is a prerequisite for cell cycle progression [15]. We aimed, in our study, to examine expression and evaluate the diagnostic utility of CK8, CK10, CK13, and CK17 in benign lesions, laryngeal dysplasia, and laryngeal SCC.

## 2. Materials and Methods

### 2.1. Patient Data

The Ethics Committee of the Medical Faculty, University of Belgrade (ethics committee decision: 1322/IV-11) approved this retrospective study. Archives of the Pathology Department in the University Clinical Center of Serbia were searched for cases diagnosed in the period from 1 January 2010 to 31 December 2020 as SCC of the larynx, laryngeal papilloma, laryngeal polyp, and laryngeal dysplasia. Patient histories were searched for information regarding age, sex, presence of metastasis in lymph nodes (N stage in TNM classification), number of lymph nodes with metastatic disease, and prior chemo/radiotherapy. Patients who were not treated with radio/chemotherapy and had available information about lymph node status were included in the study. Representative paraffin blocks and corresponding slides stained with hematoxylin and eosin (H&E) were gathered. Two experienced pathologists reviewed the slides. Specimens that contained normal laryngeal epithelia and/or glandular structures were selected for the study. Corresponding paraffin blocks were further evaluated for their suitability for further processing. One-hundred and fifty-one cases were obtained and divided into three subgroups. In the first subgroup, there were 83 (55.0%) cases with SCC of the larynx, from which 44 patients underwent complete laryngectomy and 39 patients underwent surgical removal of vocal cords.

Each larynx obtained from a complete laryngectomy was decalcified after fixation in buffered formalin. Decalcification was performed by placing the larynx into 3% formic acid. The decalcifying solution was changed daily until the laryngeal tissue was able to be appropriately cut. Decalcification did not affect IHC staining. There was no difference in IHC staining between normal and tumoral tissue obtained by laryngectomy and previously decalcified, and the tissue processed without decalcification. In the second subgroup, 28 (18.4%) cases of laryngeal dysplasia were included. The third group, with benign lesions, consisted of 40 (26.5%) patients, from which 20 (12.3%) had a diagnosis of a laryngeal polyp and 20 (12.3%) had a diagnosis of laryngeal papilloma.

### 2.2. Immunohistochemistry

Immunostaining was standardized using an appropriate positive control for each antibody, according to manufacturer instructions. A negative control was performed, omitting the primary antibody, following standard staining procedure. Immunohistochemistry (IHC) was carried out on 5 μm sections using specific monoclonal antibodies against CK8 (Dako mouse anti-cytokeratin 8, clone35βH11, dilution 1:50), CK10 (Thermo Scientific monoclonal mouse anti-cytokeratin 10, clone DE-K10, dilution 1:400), CK13 (BioSB monoclonal rabbit anti-cytokeratin 13 clone EP69, dilution 1:50), and CK17 (DAKO monoclonal mouse anti-cytokeratin 17 clone E3, dilution 1:20). The staining procedure was performed manually, according to manufacturer instructions. For all antibody external positive controls, skin sections with adnexal structures (hair follicles, sebaceous and sweat glands) were used. For all internal positive controls, fragments of normal laryngeal epithelium with laryngeal glands and myoepithelial cells were used. Cytoplasmic and/or membranous positivity was considered a positive result.

A semi-quantitative approach was used as a scoring method. Immunoreactivity of antibodies was graded as follows for CK8, CK10, and CK17: 0—without expression; 1—≤5% positive cells; 2—5% to ≤50% positive cells; 3—>50% to ≤75% positive cells; 4—>75% positive cells. The intensity of staining ranged from weak to strong but was not considered in the study due to known problems in the standardization of IHC staining and interobserver disagreement.

Immunoreactivity of the CK13 antibody, whose expression is lost in laryngeal carcinomas, was graded as follows: 0—100% positive cells (except basal cells, which are negative in normal epithelium); 1—>75% positive cells; 2—> 50% to ≤75% positive cells; 3—>5% to ≤50% positive cells and 4—≤5% of positive cells. The loss of intensity of staining ranged from weak to strong but was not considered in the study due to known problems in the standardization of IHC staining and interobserver disagreement. Percentages of stained cells were estimated on the whole surface of the lesion. Two experienced pathologists independently evaluated all immunohistochemically stained slides and scored the immunoreactivity of all used antibodies. For cases in which there was disagreement between observers, a consensus was obtained.

### 2.3. Statistical Analysis

Results are presented as count (%) or means ± standard deviation depending on data type and distribution. To assess the correlation between variables, Spearman rank correlation was used. Diagnostic characteristics of examined immunohistochemical markers were analyzed using Receiver Operating Characteristics (area under the curve, sensitivity, specificity). All *p*-values less than 0.05 were considered significant. All data were analyzed using SPSS 20.0 (IBM Corp. Released 2011. IBM SPSS Statistics for Windows, Version 20.0. Armonk, NY, USA: IBM Corp.).

## 3. Results

### 3.1. Case Characteristics

The distribution of patient baseline characteristics, diagnosis, dysplasia grade, tumor grade, tumor stage, invasion depth, lymph node metastasis, and N stage are presented in Table 1. The sample included 151 patients, 126 males, and 25 females. The average age was 60.3 ± 11.0 years, ranging from 31 to 82, 60.8 ± 10.6 for males, ranging from 31 to 82, and 57.9 ± 13.2 for females, ranging from 36 to 81.

### 3.2. Immunohistochemical Findings in the Normal Larynx

No expression of CK8 and CK10 was observed in normal laryngeal epithelia; thus, their expressions were scored as 0. We observed CK8 expression in laryngeal glandular structures. CK13 expression in normal laryngeal squamous stratified epithelia was uniform and intense, except in basal cells, in which it was negative and scored with 0. No CK17 expression was noted in normal laryngeal epithelia, so it was scored with 0.

### 3.3. Immunohistochemical Findings in Benign Lesions

There was no CK8 expression in benign lesions. CK10 expression was noted in 10 (25.0%) cases. Loss of CK13 expression was observed in 7/40 (17.5%) cases of benign lesions. Expression of CK17 was noted in 22/40 (55.0%) cases (Table 2).

#### 3.3.1. Immunohistochemical Findings in Laryngeal Polyps

There was no CK8 expression in our cases of laryngeal polyps (Figure 1B). CK10 positivity was seen in 10/20 (50%) of our cases, with a positivity score ranging from 1 to 3. Suprabasal layers were CK10-positive, and positivity was focal and/or patchy (Figure 2C). Diffuse CK10 positivity was observed in only one case. In 6/20 (30.0%) cases, a loss of CK13 expression was noted. Loss of CK13 expression was patchy, while the rest of the epithelium showed diffuse and intense positivity. In 14/20 (70%) cases, CK13 positivity was uniform and strong (Figure 1D). No dysplastic changes were observed in foci with loss of CK13 expression. CK17 expression was noted in 19/20 (95.0%) selected cases (Figure 1E). Positivity varied from patchy to diffuse. We did not notice any morphological changes (architectural or cellular atypia) or inflammation in CK17-positive parts of laryngeal polyps.

#### 3.3.2. Immunohistochemical Findings in Laryngeal Papilloma

No expression of CK8 (Figure 2B) and CK10 (Figure 2C) was detected in our cases. CK13 expression was lost focally in 1/20 (5.0%) cases, while there were no dysplastic changes in that focus. In 19/20 (95%) cases, CK13 positivity was uniform and strong (Figure 2D). CK17 expression was observed in 3/20 (15.0%) cases. CK17 was expressed in suprabasal cells. Positivity was focal in one case (5.0%), positivity was patchy in four cases (20%), and in one case (5%), large areas were diffusely positive. In CK17-positive areas, there were no morphological findings of dysplasia. In 17/20 (85%) of the papilloma cases, no CK17 positivity was observed (Figure 2E).

### 3.4. Immunohistochemical Findings in Laryngeal Dysplasia

There was no CK8 expression in our dysplasia cases. CK10 expression was detected in 20/28 (71.4%) cases, and loss of CK13 expression was noted in all cases. Loss of CK13 positivity varied from patchy to diffuse in the dysplastic epithelium. In one case (3.6%), loss of expression in less than 25% of cells was observed (score 1), in seven cases (25%), CK13 expression was lost in 25–50% of cells in the lesion (score 2), in sixteen cases (57.1%), CK13 expression was lost in 50–95% of cells (score 3), and in four cases (14.3%), loss of CK13 expression was noted in more than 95% of cells in the lesion (score 4). CK17 expression was detected in 15/28 (53.6%) of the cases of laryngeal dysplasia (Table 2).

#### 3.4.1. Immunohistochemical Findings in Low-Grade Dysplasia

CK10 positivity was observed in 9/10 (90.0%) of the low-grade dysplasia cases (Figure 3C), with positivity scores ranging from 1 to 4. CK10 expression was noted in suprabasal layers and varied from focal to patchy or diffuse. Multiple expression patterns were observed in each lesion with CK10 expression. CK13 expression was lost in all of the cases (Figure 3D). In 3/10 (30%) of our low-grade dysplasia cases, CK17 expression was noted. No CK17 expression was noted in 7/10 (70%) of the low-dysplasia cases (Figure 3E).

#### 3.4.2. Immunohistochemical Findings in High-Grade Dysplasia

In our sample of high-grade dysplasia, 11/18 (61.2%) cases were CK10-positive, while 7/18 (38.8%) cases were CK10-negative (Figure 4C). CK13 expression was lost in all dysplasia cases in areas with dysplastic morphological changes (Figure 4D). CK17 expression was observed in 12/18 (66.7%) cases of high-grade dysplasia (Figure 4E). The intensity of expression varied from weak to strong, and multiple expression patterns were observed in each lesion. In two cases of high-grade dysplasia, there was a strong diffuse pattern (score 4). (Table 2)

### 3.5. Immunohistochemical Findings in Laryngeal SCC

CK8 expression was found in 13/83 (15.7%) of our cases (Table 2). Expression was weak and focal, except in one case with diffuse and strong CK8 positivity. No CK8 expression was observed in 70/83 (84.3%) of the SCC cases (Figure 5B). Focal CK10 expression was observed in 28/83 (33.7%) cases in the proximity of foci of keratinization, while in 55/83 (65.3%) cases, CK10 expression was not found (Figure 5C). Loss of CK13 immunoreactivity was detected in all selected cases (Figure 5D). Complete loss of CK13 expression was detected in 36/83 (46.3%) of our cases, while in other cases, foci of retained CK13 expression were noted, mainly around foci of keratinization. Overall, 79/83 (95.2%) of the cases were CK17-positive (Figure 5E). Positivity was focal, patchy, or diffuse. Different patterns of expression were observed in the same lesion. Focal positivity, diffuse CK17-positive areas, and areas with no CK17 expression were observed in the same lesion.

Analysis of interobserver agreement in evaluating CK8, CK10, CK13, and CK17 expression yielded weighted kappa values of 1 (95% CI 0.996–1.004; *p* < 0.001), 0.96 (95% CI 0.954–0.961; *p* < 0.001), 0.97 (95% CI 0.995–0.971; *p* < 0.001), and 0.93 (95% CI 0.924–0.929; *p* < 0.001), respectively.

To assess the diagnostic value of CK8, CK10, CK13, and CK17 for distinguishing benign lesions from SCC, laryngeal dysplasia from benign lesions, laryngeal dysplasia from SCC, and high-grade dysplasia from low-grade dysplasia, ROC curves were plotted (Figure 6).

For CK13 expression, we determined a cutoff value of ≥1 (sensitivity 1.00, specificity 0.825, AUC 0.975) for the diagnosis of SCC in the differential diagnosis between SCC and benign lesions (Table 3). For CK13 expression, a cutoff value of ≥1 was chosen (sensitivity 1.00, specificity 0.825, and AUC 0.979) in favor of a laryngeal dysplasia diagnosis in distinguishing between dysplastic and benign lesions (Table 3). For a diagnosis of laryngeal SCC vs. dysplasia, a cutoff value of ≥2 (sensitivity 0.783, specificity 0.571, AUC 0.739) for CK17 expression was determined (Table 3). For the differential diagnosis between low-grade and high-grade dysplasia, a cutoff value of ≥2 (sensitivity 0.800, specificity 0.611, AUC 0.742) for CK10 expression was chosen for the diagnosis of low-grade dysplasia (Table 3).

## 4. Discussion

We noted the loss of CK13 expression in 7/40 (17.5%) of benign lesions, while CK13 expression was lost in all cases of laryngeal dysplasia and laryngeal SCC. It was thus concluded that CK13 is a useful marker for diagnosing lesions with malignant potential. We noted that CK17 expression indicates lesions with higher malignant potential: high-grade dysplasia and SCC of the larynx. We observed an increase in CK10 expression in dysplasias compared to benign lesions, while CK10 expression was decreased in SCC compared to benign lesions. CK10 expression was observed in 90% of our cases of low-grade dysplasia, and we noted that CK10 expression was lower in lesions with higher malignant potential: 61% of the high-grade dysplasia cases and 33.7% of the SCC cases. CK8 expression was noted only in the laryngeal SCC cases. However, only 13 (15.7%) of the SCC cases were positive for CK8, and statistical analyses showed that CK8 is not a reliable diagnostic marker for laryngeal SCC.

Using ROC analysis, we determined a cutoff value of ≥1 (sensitivity 1.00, specificity 0.825, AUC 0.975) for CK13 expression, shown as a highly sensitive and highly specific diagnostic tool for distinguishing epithelium in benign lesions from laryngeal SCC. We also chose a cutoff value of ≥1 (sensitivity 1.00, specificity 0.825, and AUC 0.979) for CK13 expression as a highly sensitive diagnostic and highly specific tool for discriminating between laryngeal dysplasia and benign lesions. If there is any loss of CK13 expression in a lesion, the lesion is dysplasia or SCC, depending on morphological findings. CK13 can be used as a marker to determine the status of resection lines. CK13 expression is homogeneous and intense in normal laryngeal epithelium, so any loss or decrease in expression is easily observed. If there is a reduced intensity or inhomogeneous pattern of CK13 expression on the resection lines, the lesion should be considered on the resection line. Loss of CK13 expression, however, did not prove to be a significant diagnostic marker for the differential diagnosis of high-grade and low-grade dysplasia, or for differential diagnosis of SCC and dysplasia.

Loss of CK13 expression has been observed in dysplasias and SCC of the oral region [17,18,19], uterine cervix [20], and differentiated vulvar intraepithelial neoplasia (dVIN) [21], which is in agreement with our study. To our knowledge, immunohistochemical expression of CK13 was not studied in laryngeal lesions.

Several studies have provided a partial explanation for the loss of CK13 expression. Deletion of the CK13 gene through LOH analysis was detected in 72 cases of laryngeal SCC. It is presumed that CK13 acts as a tumor suppressor gene [8]. One cDNA microarray-based study showed that loss of expression of CK13 and CK4 is an essential feature of oral epithelial dysplasia and oral squamous cell carcinoma (OSCC) [22]. Another study using cell culture concluded that epigenetic alterations—transcriptional silencing, post-transcriptional repression, hypermethylation of the CK13 promoter, and altered methylation of histone H3 in the CK13 promoter—have important roles in the downregulation of CK13 in OSCC [23].

Statistical analysis showed that CK17 expression indicates the diagnosis of laryngeal SCC. In the differential diagnostic dilemma between laryngeal SCC and laryngeal dysplasia, CK17 can be used as a sensitive immunohistochemical marker for a determined cutoff value of ≥2 (sensitivity 0.783, specificity 0.571, AUC 0.739) in favor of the diagnosis of SCC. If more than 5% of cells in the lesion are CK17-positive, a diagnosis of laryngeal SCC should be favored. We concluded that CK17 is not a useful marker for distinguishing benign lesions from laryngeal SCC or laryngeal dysplasia. Our results differ from the results obtained in only two previous studies that studied CK17 expression in laryngeal SCC [6,7]. The differences in the results are a consequence of the fact that the mentioned studies were performed on smaller sample sizes of laryngeal SCC and that the previous studies either did not include benign lesions [7] or included only a small number of benign lesions [6]. A different interpretation of CK17 positivity also contributed to the differences in results. We concluded that CK17 is not a reliable marker for determining the status of resection lines, which differs from previous studies [6]. CK17 expression is not constant and uniform in lesions with malignant potential. In the same lesion, there are foci without CK17 expression, foci with low CK17 expression in single scattered cells, and foci of homogeneous expression of high intensity. For determining the status of resection lines, a marker such as CK13 with a uniform expression pattern is more suitable.

The diagnostic significance of CK17 expression was examined in oral mucosal dysplasias and SCC of the oral region, where CK17 positivity was observed in 32.4% [24], 67% [25], and 74% [26] of cases of oral mucosa dysplasia. In our study, 53.6% of dysplasia cases were CK17-positive, which is consistent with previous studies [24,25,26]. A higher percentage of CK17-positive dysplasia was found in studies with larger sample sizes than ours. In previous studies, 96.2% [17] and 90.4% [19] of oral SCC were CK17-positive. Our results with 95.2% of CK17-positive cases were very similar.

In previous studies, CK17 expression has been observed in SCC of oral and oropharyngeal regions, esophagus [27], uterine cervix [20,28], and anal region [29]. CK17 expression is observed in malignant tumors that do not arise from squamous cell epithelium. It also is observed in thyroid papillary carcinoma [30], colorectal adenocarcinoma [31], pancreatic carcinoma [12], non-small cell lung carcinoma [32], and carcinomas of the ovary [33].

We observed an increase in CK10 expression in dysplasias compared to benign lesions. CK10 expression was observed in 90% of our cases of low-grade dysplasia. In contrast, CK10 expression was decreased in high-grade dysplasias and SCC compared to low-grade dysplasia cases. (Table 2). For a chosen cutoff value of ≥1, CK 10 can be used as a moderately specific and sensitive marker (sensitivity 0.714, specificity 0.750, AUC 0.762) in the differential diagnosis between dysplasia and benign lesions. If there is any CK10 positivity in lesion, a diagnosis of laryngeal dysplasia should be considered. For the chosen cutoff value of ≥2, CK10 is a highly sensitive, but moderately specific (sensitivity 0.800, specificity 0.611, AUC 0.742) marker in the differential diagnosis between low-grade and high-grade dysplasia. If there is CK10 expression in more than 5% of cells in a lesion, the diagnosis of low-grade dysplasia should be favored. To our knowledge, the expression of CK10 has not been studied in laryngeal dysplasia. mRNA expression in laryngeal SCC has been tested for diagnostic purposes in only one study, where the authors concluded that expression of CK10 is a highly sensitive (0.909) and moderately specific (0.692) diagnostic tool for diagnosing laryngeal SCC (9), which disagreed with our results. The difference in results may be a consequence of the mRNA extraction methodology because mRNA was extracted from all cells in the sample rather than specifically selecting SCC cells. Previous studies were conducted on cervical lesions [20] and skin lesions [34], where the loss of CK10 expression was observed as a sign of malignant transformation. These findings are compatible with our results.

CK8 was expressed in only 13 (15.7%) of our cases of laryngeal SCC, while we did not notice CK8 expression in our benign lesions or in cases of laryngeal dysplasia. However, we concluded that CK8 is not a sensitive, but is a specific diagnostic marker for laryngeal SCC. In previous studies conducted on laryngeal SCC [9], cervical lesions [20,35], and oral lesions [36], it was observed that CK8 was expressed only in SCC of the studied regions, which agrees with our results. Studies conducted on cervical lesions showed that CK8 is a reliable diagnostic marker for cervical SCC [20,36], which is in disagreement with our study. This disagreement is probably a consequence of the different process of tumorigenesis in cervical and laryngeal SCC.

The above-mentioned and analyzed characteristics of immunohistochemical marker expression can be transformed into a decision tree that can help pathologists discriminate between laryngeal SCC, laryngeal dysplasia, and benign lesions. The proposed decision tree is shown in Figure 7.

### Strengths and Limitations of the Present Study

This is the first study in which the expression of CK8, CK10, CD13, and CK17 was examined for diagnostic purposes in laryngeal benign and malignant lesions. However, our sample is limited, with a small number of laryngeal dysplasia samples, which can limit the generalization of results. Future studies should include larger samples of high-grade and low-grade laryngeal dysplasia cases. Different clones of the same anti-bodies should be used to determine which clone is most suitable for diagnostics.

## 5. Conclusions

Loss of CK13 expression is the most reliable diagnostic tool for diagnosing laryngeal lesions with malignant potential. Loss of CK13 expression is also a very reliable diagnostic tool for determining resection lines. In lesions with diminished CK13 expression, CK17 could be used as an auxiliary immunohistochemical marker in diagnosing laryngeal SCC. In CK13-negative and CK17-positive lesions, differentiating between high-grade and low-grade dysplasia, CK10 positivity could be used to determine low-grade dysplasia. CK8 is not a significant IHC marker for the differential diagnosis between benign laryngeal lesions, laryngeal dysplasia, and laryngeal SCC.

## Figures and Tables

**Figure 1 diagnostics-12-03203-f001:**
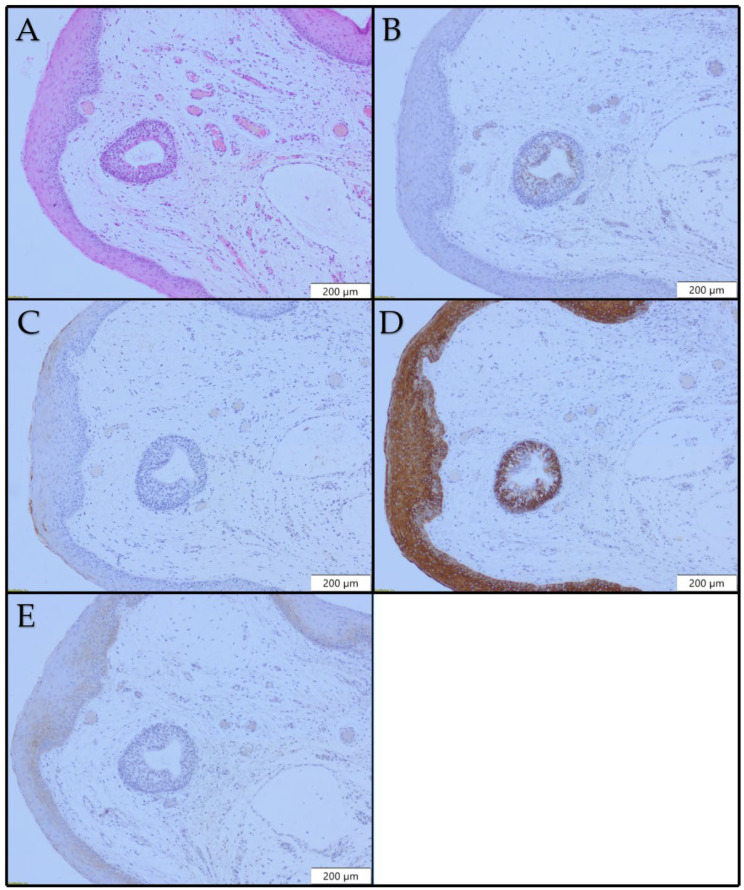
Histopathological and immunohistochemical profiles of laryngeal polyp ×100: (**A**) H&E; (**B**) no CK8 expression; (**C**) focal CK10 expression; (**D**) uniform CK13 expression; (**E**) CK17 expression.

**Figure 2 diagnostics-12-03203-f002:**
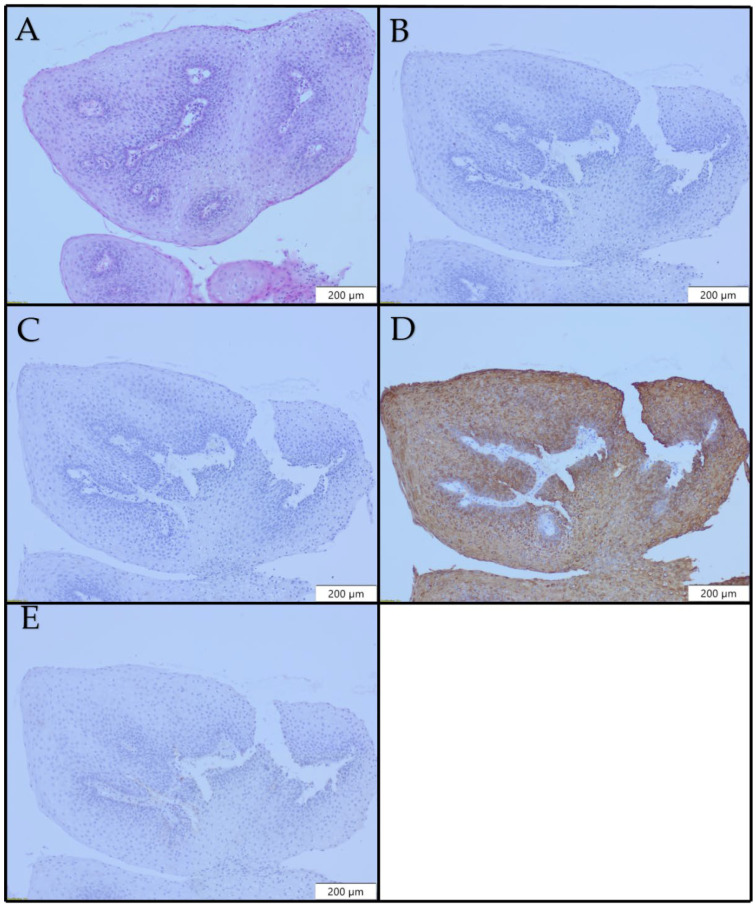
Histopathological and immunohistochemical profiles of laryngeal papilloma ×100: (**A**) H&E; (**B**) no CK8 expression; (**C**) focal CK10 expression; (**D**) uniform CK13 expression; (**E**) no CK17 expression.

**Figure 3 diagnostics-12-03203-f003:**
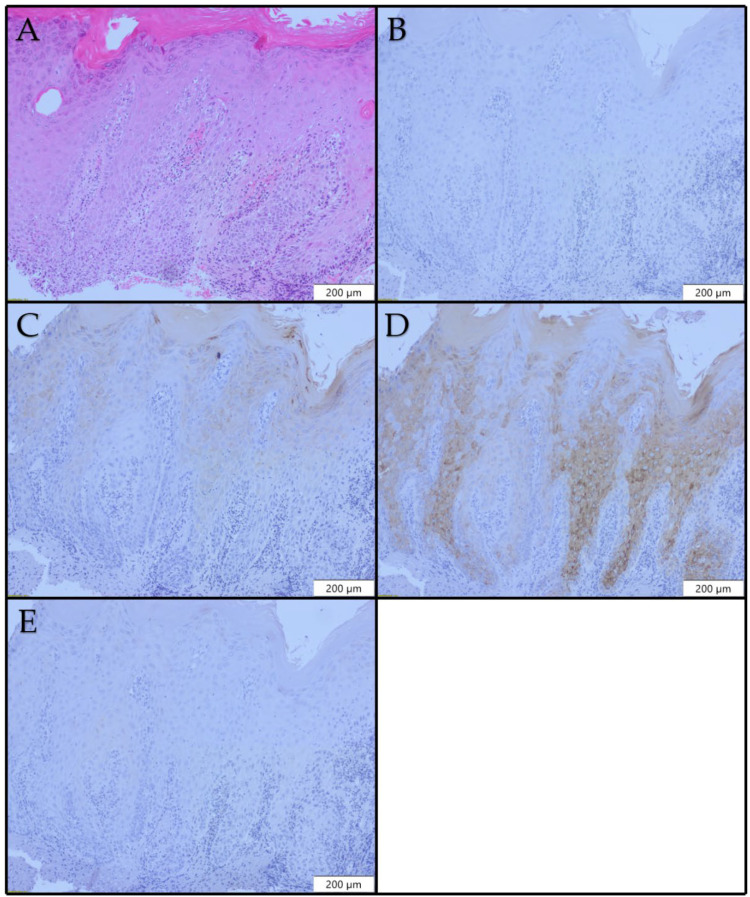
Histopathological and immunohistochemical profiles of laryngeal low-grade dysplasia ×100: (**A**) H&E; (**B**) no CK8 expression; (**C**) areas of CK10 expression; (**D**) patchy loss of CK13 expression; (**E**) no CK17 expression.

**Figure 4 diagnostics-12-03203-f004:**
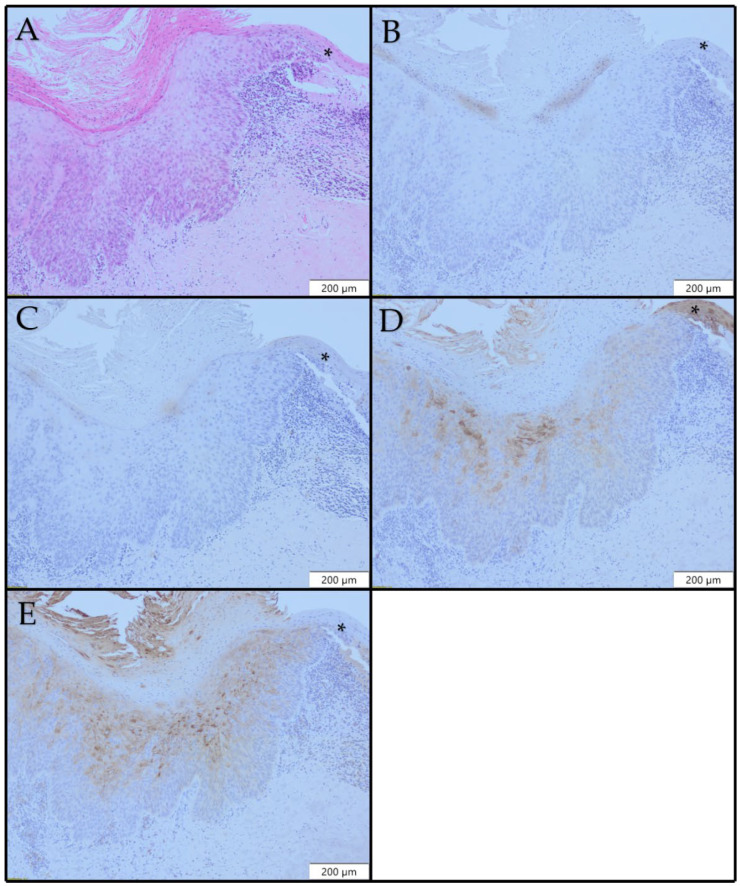
Histopathological and immunohistochemical profiles of laryngeal high-grade dysplasia ×100: (**A**) H&E; (**B**) no CK8 expression; (**C**) no CK10 expression; (**D**) patchy loss of CK13 expression; (**E**) patchy CK17 expression in high-grade dysplasia. (A*) H&E staining, fragment of normal laryngeal epithelium; (B*) no CK8 expression in normal epithelium, (C*) no CK10 expression in normal epithelium, (D*) Diffuse and uniform expression of CK13 in normal epithelium, (E*) no CK17 expression in normal epithelium.

**Figure 5 diagnostics-12-03203-f005:**
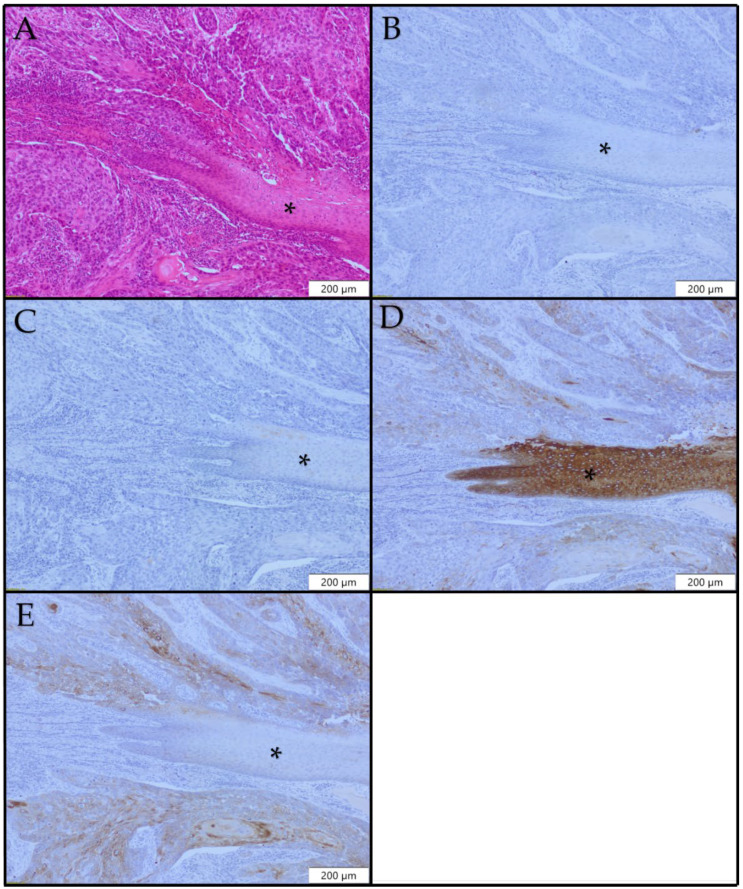
Histopathological and immunohistochemical profiles of laryngeal SCC ×100: (**A**) H&E; (**B**) no CK8 expression; (**C**) no CK10 expression (**D**) diffuse loss of CK13 expression; (**E**) diffuse CK17 expression. (A*) H&E staining, fragment of normal laryngeal epithelium; (B*) no CK8 expression in normal epithelium; (C*) no CK10 expression in normal epithelium; (D*) diffuse and uniform expression of CK13 in normal epithelium; (E*) no CK17 expression in normal epithelium.

**Figure 6 diagnostics-12-03203-f006:**
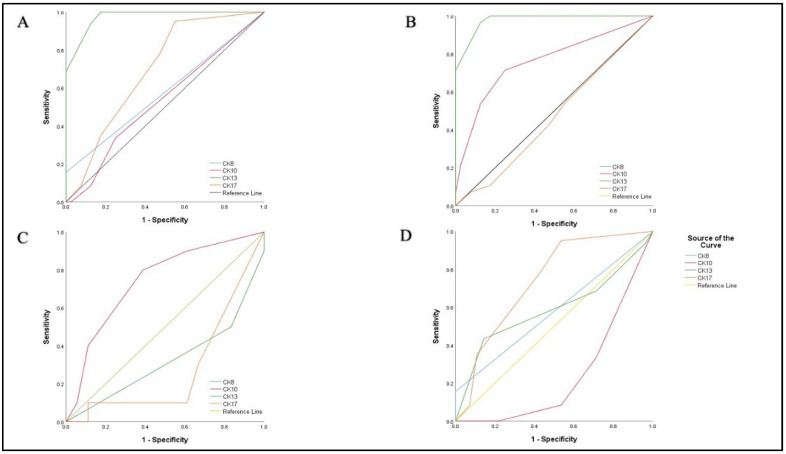
Receiver operating characteristic curves of the logistic regression models. (**A**) CK8, CK10, CK13, CK17 in differential diagnosis between laryngeal SCC and benign lesions. (**B**) CK8, CK10, CK13, CK17 in differential diagnosis between laryngeal dysplasia and benign lesions. (**C**) CK8, CK10, CK13, CK17 in differential diagnosis between laryngeal SCC and laryngeal dysplasia. (**D**) CK8, CK10, CK13, CK17 in differential diagnosis between low-grade and high-grade dysplasia.

**Figure 7 diagnostics-12-03203-f007:**
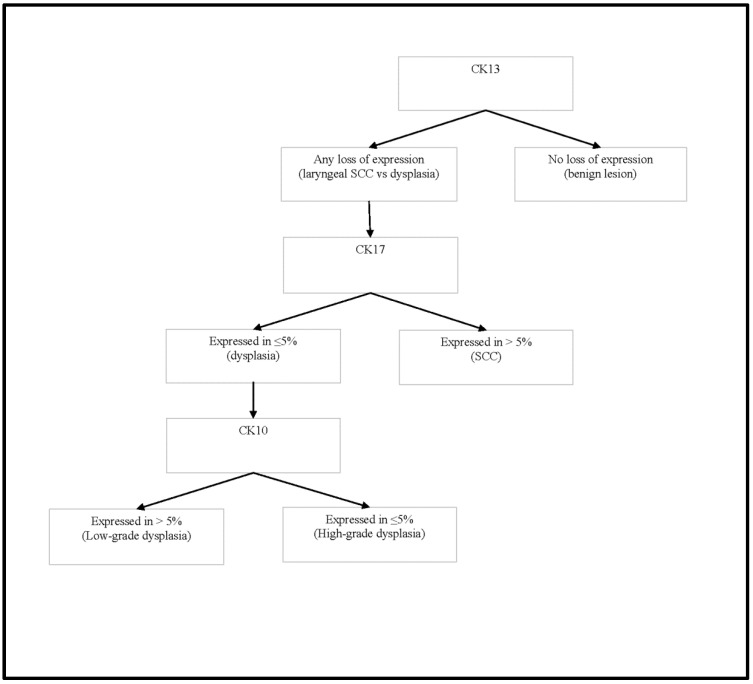
Cytokeratin expression decision tree.

**Table 1 diagnostics-12-03203-t001:** Baseline characteristics of patients.

	All Benign n = 40		Laryngeal Papilloma n = 20		Laryngeal Polyp n = 20		All Dysplasia n = 28		Low-Grade Laryngeal Dysplasia n = 10		High-Grade Laryngeal Dysplasia n = 18		Laryngeal Carcinoma n = 83	
Mean age (yrs) ±SD	51.0 ± 11.7		51.7 ± 14.0		50.4 ± 9.2		64.4 ± 8.6		64.0 ± 8.8		64.0 ± 8.5		64.4 ± 8.7	
Parameters	N	%	N	%	N	%	N	%	N	%	N	%	N	%
Sex														
Male	30	75.0	15	75.0	15	75.0	24	85.7	7	70.0	17	94.4	73	87.9
Female	10	25.0	5	25.0	5	25.0	4	14.3	3	30.0	1	5.6	10	12.1
Tumor grade														
G1													20	24.1
G2													54	65.1
G3													9	10.8
Tumor stage														
pT1													39	47.0
pT2													7	8.4
pT3													23	27.7
pT4													14	16.9
Lymph node														
N0													57	68.7
N1													6	7.2
N2													7	8.4
N3													13	15.7

**Table 2 diagnostics-12-03203-t002:** Immunohistochemical expression of CK8, CK10, CK13, and CK17 in benign lesions, laryngeal dysplasia, and laryngeal SCC.

Benign Lesions							Laryngeal Dysplasia						Malignant Lesions	
IHC Score	Polyp		Papilloma		All Benign Lesions		Low-Grade		High-Grade		All Dysplasia Cases		Laryngeal SCC	
CK8	N	%	N	%	N	%	N	%	N	%	N	%	N	%
0	20	100.0	20	100.0	40	0.0	10	0.0	18	100.0	28	100.0	70	84.3
1	0	0.0	0	0.0	0	0.0	0	0.0	0	0.0	0	0.0	8	9.6
2	0	0.0	0	0.0	0	0.0	0	0.0	0	0.0	0	0.0	4	4.8
3	0	0.0	0	0.0	0	0.0	0	0.0	0	0.0	0	0.0	0	0.0
4	0	0.0	0	0.0	0	0.0	0	0.0	0	0.0	0	0.0	1	1.3
CK10														
0	10	50.0	20	100.0	30	75.0	1	10.0	7	38.8	8	28.6	55	66.3
1	5	25.0	0	0.0	5	12.5	1	10.0	4	22.2	5	17.9	21	25.3
2	4	20.0	0	0.0	4	10.0	4	40.0	5	27.8	9	32.1	7	8.4
3	1	5.0	0	0.0	1	2.5	3	30.0	1	5.6	4	14.3	0	0.0
4	0	0.0	0	0.0	0	0	0	0.0	1	5.6	2	7.1	0	0.0
CK13														
0	14	70.0	19	95.0	33	82.5	0	0.0	0	0.0	0	0.0	0	0
1	2	10.0	0	0.0	2	5	1	10.0	0	0.0	1	3.6	5	6
2	4	20.0	1	5.0	5	12.5	4	40.0	3	16.7	7	25.0	21	25.3
3	0	0.0	0	0	0	0	4	40.0	12	66.6	16	57.1	21	25.3
4	0	0.0	0	0	0	0	1	10.0	3	16.7	4	14.3	36	43.4
CK17														
0	1	5.0	17	85.0	18	45.0	7	70.0	6	33.3	13	46.4	4	4.8
1	2	10.0	1	5.0	3	7.5	2	20.0	1	5.6	3	10.7	14	16.9
2	11	55.0	1	5.0	12	30.0	0	0.0	9	50.0	9	32.1	36	43.4
3	3	15.0	1	5.0	4	10.0	1	10.0	0	0.0	1	3.6	22	26.5
4	3	15.0	0	0.0	3	7.5	0	0.0	2	11.1	2	7.2	7	8.4

**Table 3 diagnostics-12-03203-t003:** Diagnostic values of tests based on ROC analyses.

	Area (95% CI)	*p*-Value	Cutoff	Sensitivity	Specificity	PPV	NV
Laryngeal SCC vs. Benign lesions							
CK8	0.578 (0.476–0.681)	0.160	≥1	0.157	1.000		
CK10	0.532 (0.422–0.632)	0.565	≥1	0.337	0.750		
CK13	0.975 (0.952–0.999)	<0.001	≥1	1.000	0.825	0.92	1.00
CK17	0.699 (0.591–0.807)	<0.001	≥1	0.952	0.450	0.82	0.85
Laryngeal dysplasia vs. Benign lesions							
CK8	0.500 (0.360–0.640)	1.000	≥1	0.000	0.000		
CK10	0.762 (0.641–0.883)	<0.001	≥1	0.714	0.750	0.67	0.79
CK13	0.979 (0.954–1.000)	<0.001	≥1	1.000	0.825	0.80	1.00
CK17	0.474 (0.335–0.613)	0.713	≥1	0.536	0.450		
Low-grade vs. High-grade dysplasia							
CK8	0.500 (0.273–0.727)	1.000	≥1	0	0		
CK10	0.742 (0.548–0.935)	0.037	≥2	0.800	0.611	0.53	0.85
CK13	0.325(0.103–0.547)	0.131					
CK17	0.278 (0.078–0.478)	0.055					
Laryngeal SCC vs. Dysplasia							
CK8	0.578 (0.464–0.693)	0.217	≥1	0.16	1.000		
CK10	0.242 (0.125–0.359)	0.060					
CK13	0.589 (0.479–0.699)	0.160	≥3	0.43	0.86		
CK17	0.739 (0.621–0.837)	<0.001	≥2	0.783	0.571	0.84	0.47

## Data Availability

The data presented in this study are available in the article.

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
