# Peer review of "Utility of CK8, CK10, CK13, and CK17 in Differential Diagnostics of Benign Lesions, Laryngeal Dysplasia, and Laryngeal Squamous Cell Carcinoma"

_diagnostics, 2022, doi:10.3390/diagnostics12123203_

Round 1

Reviewer 1 Report

Comments are in the pdf

Reviewer 2 Report

This study aimed to evaluate the utility of cytokeratin (CK) 8, CK10, CK13, and CK17 immunohistochemistry in diagnosing laryngeal lesions, including squamous cell carcinoma (SCC). The authors performed immunohistochemistry on 151 laryngeal lesions and statistically analyzed positive scores for each marker. The authors found that decreased expression of CK13 is a reliable indicator for differentiating laryngeal SCC and dysplasia from benign lesions, CK17 is helpful in determining between laryngeal SCC and dysplasia, and CK10 is a valuable marker for distinguishing high-grade and low-grade dysplasia.

The paper is well-written, and the results are appropriately presented. To clarify some ambiguous points, the reviewer requests some revisions as follows.

1. Please define the abbreviation “IHH” (line 97) before use.

2. As for the cases of laryngeal dysplasia (line 98), the reviewer wonders whether these lesions were diagnosed on surgically removed specimens rather than biopsy specimens. Because laryngeal biopsy specimens are usually small, it may be challenging to differentiate dysplasia from SCC.

3. In Table 1, the 75.0 in the column for laryngeal papilloma, female, % is incorrectly 25.0.

4. Figures 1-4 show that the background appears to have been artificially removed. The reviewer believes that such retouching should be avoided. It would be an error that the same photomicrographs are superimposed in Figure 4E. In addition, it isn’t easy to recognize the positive signal in the overly bluish photomicrographs in Figure 4.

5. Please clarify “loss of CK13 positivity” (line 205) as a score.

6. The reviewer agrees with the authors’ opinion that CK17 does not help distinguish dysplasia and SCC from benign lesions because oral squamous epithelia with inflammation often show CK17 positivity. Please mention the histological features of the CK17-positive parts of the benign lesions. This may help in the interpretation of CK17 immunohistochemistry.

7. There are several typographical errors, such as “loss od CK13 expression” (line 222) and “cDNa” (line 334).

Round 2

Reviewer 1 Report

Revisions are to my satisfaction

Author Response

Thank you very much for your high-quality criticisms, and for your kind suggestions to improve manuscript quality, which we have tried to respond adequately.  We sincerely hope the paper will be considered for publication.

Best regards,

Novica Boricic, MD,

Institute of Pathology, Faculty of Medicine, University of Belgrade, Belgrade, Serbia;

boricic.novica@gmail.com

Reviewer 2 Report

The reviewers believe that the manuscript has been sufficiently improved for publication.

Since both “gender” (Table 1) and “sex” (line 95) are used, please consider unifying them with one or the other.

Author Response

Thank you very much for the high-quality criticism, and for your kind suggestions which we have tried to respond adequately. The answers to the present issues are as follows: 

Point 1: Since both “gender” (Table 1) and “sex” (line 95) are used, please consider unifying them with one or the other.

Response 1: Thank you for your kind suggestions. We replaced the term “gender” with “sex” in Table 1.

Best regards,

Novica Boricic, MD,

Institute of Pathology, Faculty of Medicine, University of Belgrade, Belgrade, Serbia;

boricic.novica@gmail.com